# Effect of telomere shortening on disease progression in patients with inflammatory bowel disease: A systematic review and meta-analysis protocol

**Yifan Zhang[1☯], Ze Ma[2☯], Liang Kang[3], Liu Yang[4]***

**1** School of Traditional Chinese Medicine, Beijing University of Chinese Medicine, Beijing, People's Republic of China, **2** Graduate School, Tianjin University of Traditional Chinese Medicine, Tianjin, People's Republic of China, **3** Department of Cardiovascular, Guangzhou University of Chinese Medicine, Guangzhou, People's Republic of China, **4** Department of Gastroenterology, Tianjin Academy of Traditional Chinese Medicine Affiliated Hospital, Tianjin, People's Republic of China

☯ These authors contributed equally to this work.
* 985618688@qq.com

**Data Availability Statement:** No datasets were generated or analysed during the current study. All relevant data from this study will be made available upon study completion.

## Abstract

### Introduction

Inflammatory bowel disease (IBD) remains a major public health challenge worldwide. In recent years, it has been discovered that a link between telomere shortening and disease progression in IBD patients has been present. However, there is controversy as to whether telomere shortening precipitates disease progression or disease progression causes telomere shortening. There is also a shortage of systematic reviews and data synthesis to explain the association between telomere shortening and disease progression in individuals with IBD. We aimed to systematically review the association between telomere shortening and disease advancement in individuals with IBD to inform future studies.

### Methods and analysis

We will undertake a thorough search of the electronic database from the beginning until December 31, 2023. We will search the databases: MEDLINE/PubMed, Embase, Web of Science, China National Knowledge Infrastructure (CNKI), VIP, Wanfang Database (Wanfang), CMB, Cochrane Library, Cochran Clinical Trials Registry, and the World Health Organization International Clinical Trials Registry Platform. Two reviewers will assess the discovered citations for eligibility based on the title and abstract before proceeding to the full-text and data extraction phases. These reviewers will debate and settle any conflicts that arise during the inclusion process; a third reviewer will settle any issues that remain. The validated data extraction form will be used to collect data for eligible research. The included studies will undergo a quality and bias check and will proceed meta-analysis.

### Discussion

This systematic review and meta-analysis will reveal a positive correlation between illness progression and telomere shortening in individuals with IBD, perhaps demonstrating three

**Funding:** The author(s) received no specific funding for this work.

causal links between them. This study will conduct the first systematic review and meta-analysis examining the correlation between telomere shortening and illness advancement in individuals with IBD. Exploring the connection between these two situations can enhance the comprehension of the development and advancement of IBD.

## Systematic review registration

PROSPERO registration number: CRD42024501171.

## Introduction

Inflammatory bowel disease (IBD) remains a major public health challenge worldwide. IBD consists of two main disease subtypes: Ulcerative colitis (UC) and Crohn's disease (CD). UC is a chronic non-specific inflammatory bowel disease that begins in the rectal mucosa and continues to extend, while CD is a chronic non-specific granulomatous inflammation that tends to occur in the distal ileum and cecum. Recent epidemiological surveys have shown a significant growth in the incidence of IBD in the Americas, Europe, Asia, and Africa [1]. Specifically, more than 1.5 million and 2 million people in North America and Europe are suffering from the IBD [2], and Japan's national registry system has recorded an increase in the prevalence of UC and CD (172.9 cases/105/year and 55.6/105/year, respectively) [3]. IBD is predominantly targeted at young adults [4], but the continued rise in prevalence is partly influenced by an aging population, with 10 to 15 percent of IBD diagnoses occurring in individuals over 50 years of age [5].

IBD patients is characterized by recurrent episodes, gradually shortening of the interval between attacks, and progressively severe disease. These features further augment the physical and mental challenges of patients. For IBD, while momentous theoretical advances have been made by scholars, the etiology of IBD remains largely unclear. Many studies have highlighted the critical role of the environment, genetic factors, the immune system, and the microbiota [6, 7]. It is worth noting that in the long-term course of the IBD, multiple etiological interactions drive the dysregulation of biological processes [8].

After the etiology-driven occurrence of IBD, a state of chronic low-grade inflammation permeates the disease. Chronic low-grade inflammation is a hallmark of aging and is achieved by affecting the length and function of telomeres [9, 10]. Telomere shortening, in turn, often translates into disease and aging in humans [11]. Therefore, telomere length, as a cellular biomarker of aging, can predict physical health and longevity. The available evidence suggests the presence of short telomeres in the colonic mucosa in IBD patients [12–14]. Although telomere length reduces with age in the normal colon, the rate of telomere shortening is significantly faster in IBD patients. Similarly, some studies suggest that telomere shortening due to telomeropathies serves as one of the underlying factors for the early onset of IBD. And further promote disease progression, resulting in the development of colon cancer [15].

The above research indicates an association between telomere shortening and IBD disease development, However, there is controversy as to whether telomere shortening precipitates disease progression or disease progression causes telomere shortening. There is also a shortage of systematic reviews and data synthesis to explain the connection between telomere shortening and disease advancement in individuals with IBD. Therefore, a systematic review of qualitative and quantitative meta-analyses will be done here to seek the relationship both illness progression in IBD patients and telomere shortening for future research purposes.

## Objectives

Examining and synthesizing the literature on the relationship between telomere shortening and IBD patients' illness progression is the objective of this review.

Specifically, our goals are to:

1. Summarize the literature on the correlation between telomere shortening and disease progression in patients with IBD (primary outcomes: to determine the correlation between disease progression and telomere shortening, as well as the causative relationship between the two; secondary outcome: the relationship between disease duration and telomere shortening).

2. If possible, pool studies together to carry out a meta-analysis.

3. Perform subgroup analyses according to disease subtype, age, nation, IBD treatment (getting treatment versus not getting treatment), and comorbidity.

## Methods and analysis

The protocol was created using the Preferred Reporting Items for Systematic Review and Meta-Analysis Protocols (PRISMA-P) standard [16–18]. The study has been registered with the International Prospective Register of Systematic Reviews Network (PROSPERO, CRD42024501171) as per PRISMA-P guidelines. This systematic review is exempt from ethical approval as it will solely utilize data from published literature.

## Inclusion criteria

The investigations will consist of cross-sectional observational, case-control, and cohort studies examining the relationship both disease advancement in IBD patients and telomere shortening. Adult participants (age > 18 years) with a clinical diagnosis or colonoscopically confirmed IBD (including only the UC and CD subtypes) will be included in the study. The control group consisted of adults without IBD. The primary outcome measure will determine the association between disease progression and telomere shortening, defined as the shortening of telomeres in patients with IBD due to each cell division and cellular stress compared to the normal population. As the disease progresses, the degree of telomere shortening is further determined and evaluated by quantitative polymerase chain reaction [14]. Disease progression refers to the progression of the disease from the diagnosis of IBD and is expressed in grades of mild, moderate, and severe. Assessed by clinical symptoms (e.g., frequency of bowel movements, abdominal pain), laboratory tests (e.g., erythrocyte sedimentation rate, occult blood score), and pathologic analysis [19]. Evaluations will be conducted in hospital and community settings, encompassing both inpatient and outpatient environments. All variables such as gender, nationality, language, research population, and study design were not limited.

## Exclusion criteria

Excluded studies include those that are incomplete (such as ongoing trials and preliminary results), involve animal subjects, or are in vitro studies. In the event of repeated publication of studies with the same population, preference will be given to the report with the largest sample size. In addition, conference proceedings or abstracts, reviews, editorials, and review papers will not be considered.

## Information sources and search strategy

We are going to undertake a thorough search of the electronic database without any language constraints from the beginning until December 31, 2023. We will search the databases: MEDLINE/PubMed, Embase, Web of Science, CNKI, VIP, Wanfang, CMB, Cochrane Library, Cochran Clinical Trials Registry, and the WHO International Clinical Trials Registry Platform. We will search paper databases and the reference list of included papers to discover relevant grey literature. If data is absent from the studies provided, we will connect with the authors to gain the necessary information and document any communication.

The search strategy includes the necessary keyword fields, using MeSH terms and free word searches. The search included "inflammatory bowel disease", "bowel disease, inflammatory", "Ulcerative colitis", "Crohn's disease", "telomere shortening", "telomere shortenings", "shortening, telomere", "shortenings, telomere", "disease progression", "progression, disease", "clinical progression", "progression, clinical", "disease exacerbation", "exacerbation, disease". In addition, we offer the search strategy for one database (Fig 1), and in an all-sided review, we will incorporate the full search strategy table for every database. These databases will only contain information related to human studies and will be searchable from the start of time to the present without regard to language or demographic constraints.

| Search | Actions | Details | Query | Results | Time |
|---|---|---|---|---|---|
| #10 | ... | > | Search: (((("Inflammatory Bowel Diseases"[Mesh]) OR (((bowel disease, inflammatory[Title/Abstract]) OR (Ulcerative colitis[Title/Abstract])) OR (Crohn's disease[Title/Abstract]))) AND (("Telomere Shortening"[Mesh]) OR (((telomere shortenings[Title/Abstract]) OR (shortening, telomere[Title/Abstract])) OR (shortenings, telomere[Title/Abstract])))) AND (("Disease Progression"[Mesh]) OR (((((progression, disease) OR (clinical progression)) OR (progression, clinical)) OR (disease exacerbation)) OR (exacerbation, disease))) | 8 | 04:34:05 |
| #9 | ... | > | Search: ("Disease Progression"[Mesh]) OR (((((progression, disease) OR (clinical progression)) OR (progression, clinical)) OR (disease exacerbation)) OR (exacerbation, disease)) | 624,642 | 04:31:37 |
| #8 | ... | > | Search: ((((progression, disease) OR (clinical progression)) OR (progression, clinical)) OR (disease exacerbation)) OR (exacerbation, disease) | 624,642 | 04:31:19 |
| #7 | ... | > | Search: "Disease Progression"[Mesh] Sort by: Most Recent | 208,669 | 04:30:08 |
| #6 | ... | > | Search: ("Telomere Shortening"[Mesh]) OR (((telomere shortenings[Title/Abstract]) OR (shortening, telomere[Title/Abstract])) OR (shortenings, telomere[Title/Abstract])) | 6,410 | 04:16:03 |
| #5 | ... | > | Search: ((telomere shortenings[Title/Abstract]) OR (shortening, telomere[Title/Abstract])) OR (shortenings, telomere[Title/Abstract]) | 6,307 | 04:15:30 |
| #4 | ... | > | Search: "Telomere Shortening"[Mesh] Sort by: Most Recent | 1,983 | 04:14:04 |
| #3 | ... | > | Search: ("Inflammatory Bowel Diseases"[Mesh]) OR (((bowel disease, inflammatory[Title/Abstract]) OR (Ulcerative colitis[Title/Abstract])) OR (Crohn's disease[Title/Abstract])) | 119,357 | 04:09:59 |
| #2 | ... | > | Search: ((bowel disease, inflammatory[Title/Abstract]) OR (Ulcerative colitis[Title/Abstract])) OR (Crohn's disease[Title/Abstract]) | 83,520 | 04:08:51 |
| #1 | ... | > | Search: "Inflammatory Bowel Diseases"[Mesh] Sort by: Most Recent | 99,063 | 04:07:09 |

**Fig 1. The search strategy used in the PubMed database.**

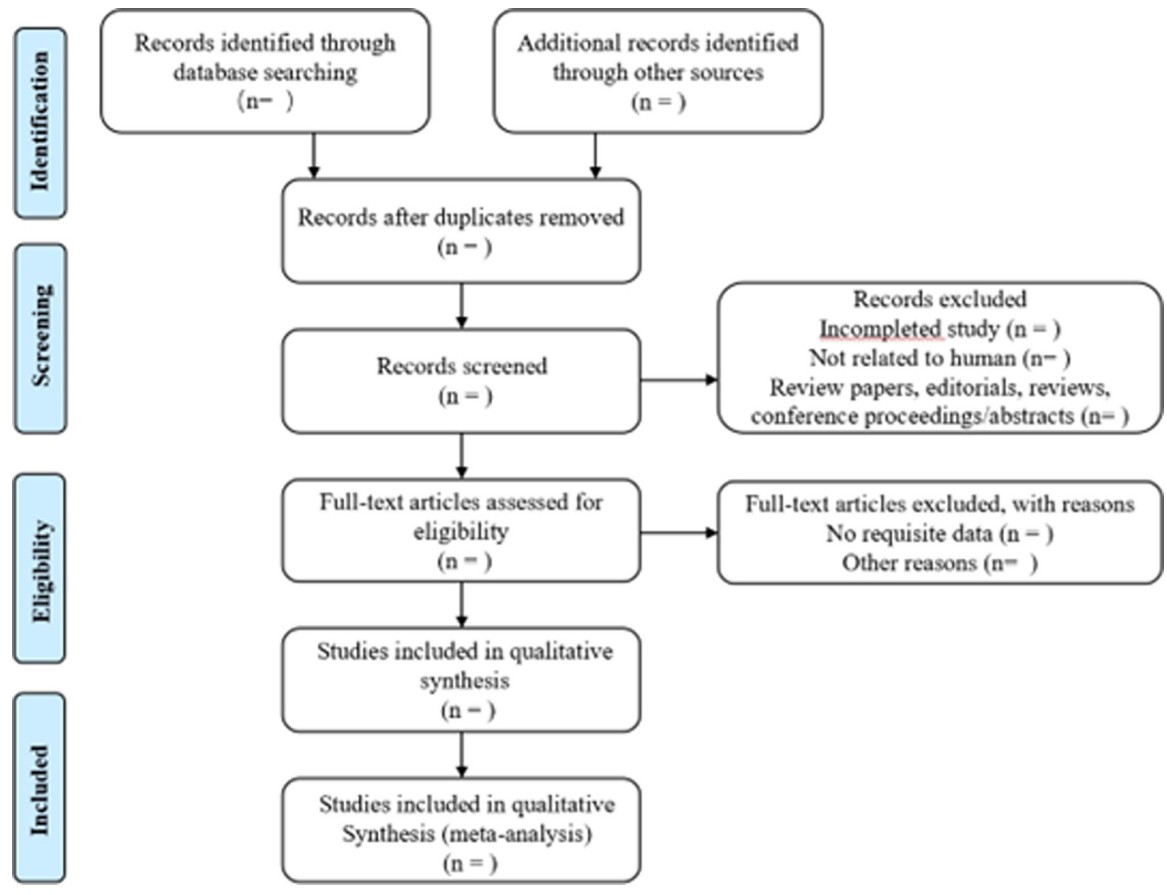

**Fig 2. Flow chat of study selection.**

## Data extraction and management

The extracted literature was assessed in three stages: title and abstract screening, full-text screening, and data extraction. This process will be carried out in duplicate. The reviewers will resolve any disputes in the included papers through thorough debate. Any lingering conflicts will be addressed and resolved by a third party.

The selection process will adhere to the PRISMA flowchart depicted in Fig 2. The information will be transferred into Review Manager 5.3 for data management.

## Data collection process

There will be two copies of the data extraction form made. For every article that has been included for data extraction, the following items will be gathered: author, year, country, gender, age (including the true age of patients in the treatment and control groups and the age of onset of the patients in the treatment group), sample size (including the total sample size and the number of individuals with IBD and the normal population), duration of disease, disease subtype (UC/CD), disease stage (mild, moderate, or severe), drug treatment, comorbidity, and telomere length (telomere shortening). If information is absent from the studies provided, we will contact the authors to gain the necessary data and document any communication. Eventually, if this information is not acquired, it will be omitted from the analysis.

### Risk of bias in individual studies

Using the proper instruments in accordance with the study design, and the risk of bias in each study that is part of the data extraction phase will be evaluated. These evaluations will also be finished by two reviewers, respectively. We will apply the Newcastle-Ottawa Scale (NOS) to assess the risk of bias [20]. If possible, we will perform sensitivity analyses for papers with low scores at risk of bias.

### Assessment of heterogeneity

We will use $\chi^2$ test and $I^2$ statistic test to assess the heterogeneity. If the results meet the $I^2$ value is <50% and the p>0.10, we aggregated the data utilizing a fixed-effect model. Otherwise, a random-effects model will be used. Expected sources of heterogeneity include disease subtype, drug use, age, country and comorbidity, and if significant heterogeneity is found, this will be flagged and subgroup analyzed for this.

### Data synthesis

Qualitative information on the identified cross-sectional observational, case-control and cohort studies will be supplied. By combining studies with comparable designs and measures, we shall do meta-analysis on primary and secondary outcomes and present the results as forest plots.

To be clear, we will only pool studies at low or moderate risk of bias. All quantitative analyses will be statistically performed using RevMan Manager 5.3 software (The Cochrane Collaboration). If a meta-analysis was not feasible due to small and highly heterogeneous studies, we performed a narrative summation of the included papers to sum up the results.

### Assessment of reporting biases

A funnel plot will be utilized to analyze for potential reporting bias whenever the number of studies exceeds 10 in the meta-analysis. Egger's test will determine the heterogeneity of the funnel plot. If p<0.05, there may be a significant reporting bias.

### Data statement

Upon completion of the review, any additional data will furnished on request.

### Confidence in cumulative evidence

Two reviewers will assess the strength of the evidence employing the Grading of Recommendations Assessment, Development, and Evaluations (GRADE) system. Evidence quality will be evaluated by considering factors such as the risk of bias, publication bias, inconsistency, indirectness, and imprecision.

### Patient and public involvement

No patient or public involvement will be included in this systematic review.

### Ethics and dissemination

We will disseminate the results of this review by publishing the manuscript in a peer-reviewed journal or by publishing the findings at relevant conferences. Ethical approval is not required for the meta-analysis proposed by the manuscript, given this is a systematic review of the already published research.

## Discussion

This study intends to appraise the correlation between telomere shortening and disease progression in individuals suffering from IBD. This systematic review and meta-analysis will reveal the correlation between IBD and telomere shortening, and may further determine the causal relationship between telomere shortening and IBD disease progression.

We speculate that there may be three possible causal relationships between the two. 1) Telomere shortening leads to the development of IBD. Telomeres are protected from natural shortening during cell division by telomerase. In telomeropathies, mutations in the telomerase reverse transcriptase lead to progressive telomere shortening [15, 21]. Telomere shortening can result in intestinal cell dysfunction and colitis by disrupting colonic barrier integrity and modulating the microbiome. 2) IBD disease progression promotes telomere shortening. Chronic inflammation and oxidative damage in the IBD colon can promote telomere shortening [22, 23]. 3) There is mutual causality between telomere shortening and IBD disease progression. Telomere shortening is not only a consequence of chronic inflammation but can be a key trigger for the inflammatory process in IBD [24, 25].

As far as we know, this study will be the first systematic review and meta-analysis exploring the correlation between telomere shortening and disease progression in individuals with IBD. The study's strengths lie in incorporation of both quantitative and qualitative analysis and its rigorous compliance with the PRISMA-P standards. The study also has certain limitations. One limitation is that UC/CD in the context of treatment, disease progression may be influenced, which will generate a deviation in revealing the causal relationship between telomere shortening and IBD disease progression. Another limitation relates to the fact that there is a lack of comprehensive information on the correlation between telomere shortening and disease advancement in IBD patients, which means that the current results may not conclusively establish a causal connection between the two. Whatever the result is, studying the association between the two will enhance our understanding of the occurrence and progression of IBD and serve as an early warning of the arrival of colon cancer.

## Supporting information

**S1 File. PRISMA-P checklist.**
(DOC)

## Author Contributions

**Conceptualization:** Yifan Zhang, Ze Ma.

**Formal analysis:** Yifan Zhang, Liang Kang.

**Funding acquisition:** Liu Yang.

**Investigation:** Ze Ma, Liang Kang.

**Methodology:** Yifan Zhang, Ze Ma.

**Supervision:** Liu Yang.

**Validation:** Ze Ma.

**Writing – original draft:** Yifan Zhang.

**Writing – review & editing:** Yifan Zhang, Ze Ma, Liu Yang.

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
