## [Decision Letter · Decision Letter 0]

9 May 2024

PONE-D-24-10905Effect of telomere shortening on disease progression in patients with inflammatory bowel disease: a systematic review and meta-analysis protocolPLOS ONE

Dear Dr. Yang,

Thank you for submitting your manuscript to PLOS ONE. After careful consideration, we feel that it has merit but does not fully meet PLOS ONE’s publication criteria as it currently stands. Therefore, we invite you to submit a revised version of the manuscript that addresses the points raised during the review process.

We look forward to receiving your revised manuscript.

Kind regards,

Mehran Rahimlou, PhD

Academic Editor

PLOS ONE

Journal Requirements:

3. Please remove your figures from within your manuscript file, leaving only the individual TIFF/EPS image files, uploaded separately. These will be automatically included in the reviewers’ PDF.

4. Please include your tables as part of your main manuscript and remove the individual files. Please note that supplementary tables (should remain/ be uploaded) as separate "supporting information" files

6. We note that this manuscript is a systematic review or meta-analysis; our author guidelines therefore require that you use PRISMA guidance to help improve reporting quality of this type of study. Please upload copies of the completed PRISMA checklist as Supporting Information with a file name “PRISMA checklist”.

Reviewers' comments:

Reviewer's Responses to Questions

**Comments to the Author**

1. Does the manuscript provide a valid rationale for the proposed study, with clearly identified and justified research questions?

Reviewer #1: Yes

Reviewer #2: Yes

2. Is the protocol technically sound and planned in a manner that will lead to a meaningful outcome and allow testing the stated hypotheses?

Reviewer #1: Partly

Reviewer #2: Yes

3. Is the methodology feasible and described in sufficient detail to allow the work to be replicable?

Reviewer #1: Yes

Reviewer #2: Yes

4. Have the authors described where all data underlying the findings will be made available when the study is complete?

Reviewer #1: Yes

Reviewer #2: No

5. Is the manuscript presented in an intelligible fashion and written in standard English?

Reviewer #1: Yes

Reviewer #2: Yes

6. Review Comments to the Author

You may also provide optional suggestions and comments to authors that they might find helpful in planning their study.

Reviewer #1: The authors present a systematic review protocol to appraise the correlation between telomere shortening and disease

progression in individuals suffering from IBD and to determine the causal relationship between telomere shortening and IBD disease progression. The topic is relevant, but there is a need to organize the protocol so that the future review can adequately respond to the objectives.

In the methods, the authors need to make some points clearer. Population, only studies involving adults?, Exposure, inflammatory bowel disease, but only ulcerative colitis and Crohn's disease? What diagnostic criteria for these diseases should have been used in the studies? Comparison, adults without inflammatory bowel disease? Outcome: telomere shortening? Disease progression?

Considering this PECO strategy, the most correct option would be to include only cross-sectional observational, case-control and cohort studies, with the tool to assess the quality and risk of bias of the studies being the NOS.

Therefore, the authors should withdraw "The investigations will consist of randomized controlled trials (RCTs) ", "We will include pertinent case studies, case series, and a qualitative report. We shall contain the most current paper relevant to the systematic review if multiple RCTs have been published under the same trial registry number.", since these types of studies will not allow establishing the causal relationship between telomere shortening and IBD disease progression.

Finally, still in the methods, the authors must define the criteria that need to be present in the selected studies and that define telomere shortening. How is this assessed in primary studies? Furthermore, they should clarify how it is defined that there has been disease progression in patients with UC/CD? If it is with clinical data, which ones and in what form? Will it need to be through pathological analysis?

In the discussion, another limitation that must be raised is that in the case of UC/CD treatment, the progression of the disease will probably be influenced, which is an important bias for a review that aims to answer the causal relationship between telomere shortening and IBD disease progression. For this hypothesis, "IBD disease progression promotes telomere shortening. Chronic inflammation and oxidative damage in the IBD colon can promote telomere shortening" the authors must consider that the treatment in observational studies represents a large bias.

Reviewer #2: Yifan Zhang et al., in this study protocol, will explore the association between telomere shortening and IBD. They propose a positive correlation, which will result in a better comprehension of the development and advancement of IBD. The overall study seems well-analyzed, provides a valid rationale for the proposed research, and has a feasible methodology. I only request that the authors specify the UC and CD definitions in the text. As a suggestion, the subgroup analysis should also include the gender and comorbidities reported.

7. PLOS authors have the option to publish the peer review history of their article (what does this mean?). If published, this will include your full peer review and any attached files.

Reviewer #1: **Yes: **Ricardo Ney Cobucci

Reviewer #2: **Yes: **Julio César Flores González

---

## [Author Response · Author response to Decision Letter 0]

17 Jun 2024

Dear Editors and Reviewers:

Thank you for your letter and for the reviewers’ comments concerning our manuscript entitled “Effect of telomere shortening on disease progression in patients with inflammatory bowel disease: a systematic review and meta-analysis protocol” (ID: PONE-D-24-10905). Those comments are valuable and helpful for revising and improving our paper, as well as the important guiding significance to our research. We have studied comments carefully and have made corrections which we hope meet with approval. Revised portions are marked in red on the manuscript. The main corrections in the paper and the responds to the reviewer’s comments are as flowing:

1. Reviewers' comments and response:

Does the manuscript provide a valid rationale for the proposed study, with clearly identified and justified research questions?

Reviewer #1: Yes

Reviewer #2: Yes

Response: 

Thank you so much.

2. Reviewers' comments and response:

Is the protocol technically sound and planned in a manner that will lead to a meaningful outcome and allow testing the stated hypotheses?

Reviewer #1: Partly

Reviewer #2: Yes

Response: 

Thanks very much. Reviewer #1 raised more detailed methodological and analytical questions in question 6, so specific improvements were responded to in question 6.

3. Reviewers' comments and response:

Is the methodology feasible and described in sufficient detail to allow the work to be replicable?

Reviewer #1: Yes

Reviewer #2: Yes

Response: 

Thank you very much.

4. Reviewers' comments and response:

Have the authors described where all data underlying the findings will be made available when the study is complete?

Reviewer #1: Yes

Reviewer #2: No

Response: 

Thanks very much. We have made a “Data statement” in the manuscript and indicated that “Upon completion of the review, any additional data will furnished on request.”

Data statement

Upon completion of the review, any additional data will furnished on request.

5. Reviewers' comments and response:

Is the manuscript presented in an intelligible fashion and written in standard English?

Reviewer #1: Yes

Reviewer #2: Yes

Response: 

Thank you so much.

6. Reviewers' comments and response:

Reviewer #1: 

The authors present a systematic review protocol to appraise the correlation between telomere shortening and disease progression in individuals suffering from IBD and to determine the causal relationship between telomere shortening and IBD disease progression. The topic is relevant, but there is a need to organize the protocol so that the future review can adequately respond to the objectives. In the methods, the authors need to make some points clearer. Population, only studies involving adults?, Exposure, inflammatory bowel disease, but only ulcerative colitis and Crohn's disease? What diagnostic criteria for these diseases should have been used in the studies? Comparison, adults without inflammatory bowel disease? Outcome: telomere shortening? Disease progression? Considering this PECO strategy, the most correct option would be to include only cross-sectional observational, case-control and cohort studies, with the tool to assess the quality and risk of bias of the studies being the NOS. Therefore, the authors should withdraw "The investigations will consist of randomized controlled trials (RCTs) ", "We will include pertinent case studies, case series, and a qualitative report. We shall contain the most current paper relevant to the systematic review if multiple RCTs have been published under the same trial registry number.", since these types of studies will not allow establishing the causal relationship between telomere shortening and IBD disease progression. Finally, still in the methods, the authors must define the criteria that need to be present in the selected studies and that define telomere shortening. How is this assessed in primary studies? Furthermore, they should clarify how it is defined that there has been disease progression in patients with UC/CD? If it is with clinical data, which ones and in what form? Will it need to be through pathological analysis? In the discussion, another limitation that must be raised is that in the case of UC/CD treatment, the progression of the disease will probably be influenced, which is an important bias for a review that aims to answer the causal relationship between telomere shortening and IBD disease progression. For this hypothesis, "IBD disease progression promotes telomere shortening. Chronic inflammation and oxidative damage in the IBD colon can promote telomere shortening" the authors must consider that the treatment in observational studies represents a large bias.

Response：

Thank you very much. According to your suggestion, we have added more specific and detailed content in the "Inclusion criteria" section, which has made our research more complete. And change the tool to assess study quality and risk of bias to NOS.

The investigations will consist of cross-sectional observational, case-control, and cohort studies examining the relationship both disease advancement in IBD patients and telomere shortening. Adult participants (age > 18 years) with a clinical diagnosis or colonoscopically confirmed IBD (including only the UC and CD subtypes) will be included in the study. The control group consisted of adults without IBD. The primary outcome measure will determine the association between disease progression and telomere shortening, defined as the shortening of telomeres in patients with IBD due to each cell division and cellular stress compared to the normal population. As the disease progresses, the degree of telomere shortening is further determined and evaluated by quantitative polymerase chain reaction [19]. Disease progression refers to the progression of the disease from the diagnosis of IBD and is expressed in grades of mild, moderate, and severe. Assessed by clinical symptoms (e.g., frequency of bowel movements, abdominal pain), laboratory tests (e.g., erythrocyte sedimentation rate, occult blood score), and pathologic analysis [20]. Evaluations will be conducted in hospital and community settings, encompassing both inpatient and outpatient environments. All variables such as gender, nationality, language, research population, and study design were not limited.

Using the proper instruments in accordance with the study design, and the risk of bias in each study that is part of the data extraction phase will be evaluated. These evaluations will also be finished by two reviewers, respectively. We will apply the Newcastle-Ottawa Scale (NOS) to assess the risk of bias [21]. If possible, we will perform sensitivity analyses for papers with low scores at risk of bias.

Reviewer #2: 

Yifan Zhang et al., in this study protocol, will explore the association between telomere shortening and IBD. They propose a positive correlation, which will result in a better comprehension of the development and advancement of IBD. The overall study seems well-analyzed, provides a valid rationale for the proposed research, and has a feasible methodology. I only request that the authors specify the UC and CD definitions in the text. As a suggestion, the subgroup analysis should also include the gender and comorbidities reported.

Response：

Thank you for your suggestion. We added definitions for UC and CD and added analyses for comorbidities to subgroup analyses.

Inflammatory bowel disease (IBD) remains a major public health challenge worldwide. IBD consists of two main disease subtypes: Ulcerative colitis (UC) and Crohn's disease (CD). UC is a chronic non-specific inflammatory bowel disease that begins in the rectal mucosa and continues to extend, while CD is a chronic non-specific granulomatous inflammation that tends to occur in the distal ileum and cecum. Recent epidemiological surveys have shown a significant growth in the incidence of IBD in the Americas, Europe, Asia, and Africa [1].

3. Perform subgroup analyses according to disease subtype, age, nation, IBD treatment (getting treatment versus not getting treatment), and comorbidity.

---

## [Decision Letter · Decision Letter 1]

24 Sep 2024

Effect of telomere shortening on disease progression in patients with inflammatory bowel disease: a systematic review and meta-analysis protocol

PONE-D-24-10905R1

Dear Dr. Liu Yang

We’re pleased to inform you that your manuscript has been judged scientifically suitable for publication and will be formally accepted for publication once it meets all outstanding technical requirements.

Kind regards,

Mehran Rahimlou, PhD

Academic Editor

PLOS ONE

Additional Editor Comments (optional):

Reviewers' comments:

Reviewer's Responses to Questions

**Comments to the Author**

1. Does the manuscript provide a valid rationale for the proposed study, with clearly identified and justified research questions?

Reviewer #1: Yes

Reviewer #3: Yes

2. Is the protocol technically sound and planned in a manner that will lead to a meaningful outcome and allow testing the stated hypotheses?

Reviewer #1: Yes

Reviewer #3: Yes

3. Is the methodology feasible and described in sufficient detail to allow the work to be replicable?

Reviewer #1: Yes

Reviewer #3: Yes

4. Have the authors described where all data underlying the findings will be made available when the study is complete?

Reviewer #1: Yes

Reviewer #3: Yes

5. Is the manuscript presented in an intelligible fashion and written in standard English?

Reviewer #1: Yes

Reviewer #3: Yes

6. Review Comments to the Author

You may also provide optional suggestions and comments to authors that they might find helpful in planning their study.

Reviewer #1: The authors met most of the reviewers' recommendations and the manuscript is ready to be published. Congratulations!

Reviewer #3: The authors addressed all the previous reviewer comments adequately and included all the requested material, data availability. It addresses well the connection between IDB and telomere shortening.

7. PLOS authors have the option to publish the peer review history of their article (what does this mean?). If published, this will include your full peer review and any attached files.

Reviewer #1: **Yes: **Ricardo Ney Cobucci

Reviewer #3: No

---

## [Editor Report · Acceptance letter]

7 Oct 2024

PONE-D-24-10905R1 

PLOS ONE

Dear Dr. Yang, 

I'm pleased to inform you that your manuscript has been deemed suitable for publication in PLOS ONE. Congratulations! Your manuscript is now being handed over to our production team.

Kind regards, 

on behalf of

Dr. Mehran Rahimlou 

Academic Editor

PLOS ONE